# Integrins and Actions of Androgen in Breast Cancer

**DOI:** 10.3390/cells12172126

**Published:** 2023-08-22

**Authors:** Chung-Che Tsai, Yu-Chen S. H. Yang, Yi-Fong Chen, Lin-Yi Huang, Yung-Ning Yang, Sheng-Yang Lee, Wen-Long Wang, Hsin-Lun Lee, Jacqueline Whang-Peng, Hung-Yun Lin, Kuan Wang

**Affiliations:** 1Graduate Institute of Cancer Biology and Drug Discovery, College of Medical Science and Technology, Taipei Medical University, Taipei 11031, Taiwan; a03130@tmu.edu.tw (C.-C.T.); slugcousin7682@tmu.edu.tw (Y.-F.C.); 2Graduate Institute of Nanomedicine and Medical Engineering, College of Medical Engineering, Taipei Medical University, Taipei 11031, Taiwan; 3Joint Biobank, Office of Human Research, Taipei Medical University, Taipei 11031, Taiwan; can_0131@tmu.edu.tw; 4Department of Pediatrics, E-DA Hospital, I-Shou University, Kaohsiung 82445, Taiwan; pedptch05680@gmail.com (L.-Y.H.); ancaly@yahoo.com.tw (Y.-N.Y.); 5School of Medicine, I-Shou University, Kaohsiung 82445, Taiwan; 6Dentistry, Wan-Fang Medical Center, Taipei Medical University, Taipei 11031, Taiwan; seanlee@tmu.edu.tw; 7School of Dentistry, College of Oral Medicine, Taipei Medical University, Taipei 11031, Taiwan; 8Department of Life Science, Fu Jen Catholic University, New Taipei City 242, Taiwan; annie88145@gmail.com; 9Department of Radiology, School of Medicine, College of Medicine, Taipei Medical University, Taipei 11031, Taiwan; 132018@h.tmu.edu.tw; 10Department of Radiation Oncology, Taipei Medical University Hospital, Taipei 110, Taiwan; 11The Ph.D. Program for Translational Medicine, College of Medical Science and Technology, Taipei Medical University and Academia Sinica, Taipei 11031, Taiwan; 12Cancer Center, Wan Fang Hospital, Taipei Medical University, Taipei 11031, Taiwan; jqwpeng@nhri.edu.tw; 13TMU Research Center of Cancer Translational Medicine, Taipei Medical University, Taipei 11031, Taiwan; 14Traditional Herbal Medicine Research Center of Taipei Medical University Hospital, Taipei Medical University, Taipei 11031, Taiwan; 15Pharmaceutical Research Institute, Albany College of Pharmacy and Health Sciences, Albany, NY 12208, USA

**Keywords:** breast cancer, androgen, androgen receptor, integrin αvβ3, PD-L

## Abstract

Androgen has been shown to regulate male physiological activities and cancer proliferation. It is used to antagonize estrogen-induced proliferative effects in breast cancer cells. However, evidence indicates that androgen can stimulate cancer cell growth in estrogen receptor (ER)-positive and ER-negative breast cancer cells via different types of receptors and different mechanisms. Androgen-induced cancer growth and metastasis link with different types of integrins. Integrin αvβ3 is predominantly expressed and activated in cancer cells and rapidly dividing endothelial cells. Programmed death-ligand 1 (PD-L1) also plays a vital role in cancer growth. The part of integrins in action with androgen in cancer cells is not fully mechanically understood. To clarify the interactions between androgen and integrin αvβ3, we carried out molecular modeling to explain the potential interactions of androgen with integrin αvβ3. The androgen-regulated mechanisms on PD-L1 and its effects were also addressed.

## 1. Introduction

Breast cancer is among the most common morbid cancers, exhibiting diverse morphological and molecular characteristics. It is commonly classified based on the presence or absence of the following hormone receptors expression (as determined by immunohistochemical analysis): estrogen receptor (ER), progesterone receptor (PR), and human epidermal growth factor receptor (HER2). Accordingly, four breast cancer subtypes are widely recognized: luminal A, luminal B, HER2-positive, and triple negative. Among these subtypes of breast cancer, triple-negative breast cancer (TNBC) has a progressive biology, resulting in a higher incidence of recurrence and metastasis, due to its special molecular phenotype which causes it to be insensitive to endocrine therapy or molecular targeted therapy [1]. TNBC is notably invasive, with around 50% of patients experiencing distant metastasis [1]. It is unresponsive to endocrine or targeted therapies, ultimately leading to tumor recurrence from residual metastatic lesions [1]. In addition to traditional nuclear receptors such as estrogen receptors (Erα and ERβ), cell surface-bound hormone receptors may contribute to normal breast development and mammary stem cells [2] and breast cancer as well [3].

Androgen, a steroid hormone, has been reported to regulate cancer progression during its receptor activation. However, the role of androgens in cancer cell progression is controversial in the specific types of cancer, including ER-positive breast cancer [4,5,6,7,8,9,10]. Compared to ER-positive breast cancer cells, androgens including dihydrotestosterone (DHT) stimulate the proliferation and migration of ER-negative breast cancer cells. This indicates that different levels of ER, PR, or HER expressed in the breast cancer cause androgen receptor (AR)-dependent or -independent cancer progression [9,10]. In addition to the well-known AR and ER, integrins have also been found to play roles in androgen functions (Table 1). Integrin-mediated cell adhesion to the extracellular matrix (ECM) is essential for normal tissue development and function. However, this adhesion process is often disrupted in cancer, leading to pathological changes. Indeed, the loss of integrin expression or the inhibition of integrin activity promotes cancer cell metastasis. Integrin-induced programmed death-ligand 1 (PD-L1) expression on cancer cells can help them avoid T cell killing, thereby enhancing cell migration and proliferation. PD-L1 binds to its receptor, PD-1, found on activated T cells and B cells, to play a vital pathogenetic role in breast cancer progression [11]. Distinct subtypes of integrins expressed on breast cancer cell membranes have been described. However, the role of the interaction of androgen and integrin in breast cancer development is not clarified. Additionally, the involvement of PD-L1 in the androgen/integrin-mediated cancer progression is still unknown. In this current review article, we have presented compelling evidence supporting the role of androgen in stimulating various biological activities in cancer cells through different receptors. Furthermore, we have highlighted the potential of PD-L1 as a therapeutic target for treating integrin-mediated AR-positive breast cancer cells.

## 2. Androgen-Induced Signal Transduction in Breast Cancer Cells

Androgen is a steroid hormone crucial in developing and maintaining male characteristics in vertebrates. The most common androgens are testosterone, DHT, and androstenedione [31]. These androgens exert their effects by binding to and activating ARs, which are present in various tissues throughout the body. Activation of ARs not only mediates a variety of physiological responses but also participates in cancer progression [32]. ARs have been found in multiple cancer cells, particularly in prostate and breast cancers [33].

The primary function of ARs is to exert their effects through direct regulation of gene transcription [34,35,36]. In human prostate cancer LNCaP cells, the activation of classical intracellular ARs by androgens regulates cell growth [35]. However, the role of the AR in breast cancer is more complex. The AR is expressed in about 60% of early breast cancer cases, with a higher prevalence in ER-positive tumors compared to ER-negative tumors. It is detected in 90% of ER-positive breast cancers and 12–50% of TNBC cases [37,38]. The activation of the AR plays a central role in regulating breast cancer progression, although its specific function in TNBC is still under debate. The signaling pathways involved in the classical intracellular androgen-induced activation of specific receptors contribute to the cell progression in breast cancer, as shown in Figure 1 and Figure 2. In ER-positive breast cancer, AR signaling acts as a counterbalance to the growth stimulatory effect of ER signaling. Additionally, biased ligands can induce AR signaling, compensating for the progression effect in both the ER-positive and ER-negative breast cancer cells through the activation of classical intracellular AR (iAR) or cell membrane AR (mAR). In molecular apocrine tumors, characterized by ER-negative and AR-positive status, androgens have an established oncogenic role [39,40]. AR signaling also promotes proliferation in ER-negative, HER2-positive breast cancer [41]. In the luminal AR subtype of TNBC, which shares similarities with molecular apocrine tumors, the AR appears to drive tumor progression [41,42]. Moreover, DHT activates the cell surface integrin αvβ3, rather than the iAR, to enhance the proliferation of TNBC cells [9,10,43].

Androgen also directly bind to the ARs in the cell membrane. Activation of this mAR induces cellular processes. Exposure of prostate cancer cells to the non-internalized testosterone albumin conjugates (TAC) regulates the actin reorganization through focal adhesion kinase (FAK)/phosphoinositide 3-kinase (PI3K) signaling. Activation of PI3K modulates the small guanosine triphosphatases cell division control protein 42 (Cdc42)/ras-related C3 botulinum toxin substrate 1 (Rac1) to secrete prostate-specific antigen (PSA). This process ultimately results in the redistribution of actin [44]. Furthermore, TAC can activate extracellular signal-regulated kinase 1/2 (ERK1/2), but not p38 mitogen-activated protein kinase (MAPK) and c-Jun N-terminal kinase (JNK), to mediate a prostate cancer cell response [45]. In melanoma A375 cells, the FAK/PI3K/Rac1/actin signaling pathway, which is induced by a different membrane opioid receptor, is also involved in regulating actin reorganization [46]. This evidence indicates that this pathway plays an important role in controlling the cell motility through various receptors. Similarly, in the ER-positive MCF-7 cells, TAC rapidly activates non-genomic FAK/PI3K/Rac1/Cdc42 signaling, triggers actin reorganization, and inhibits cell motility [47]. In addition, this mAR activation induced by TAC in combination with serum and glucocorticoid inducible kinase (SGK1) inhibition triggers pro-apoptotic responses [47]. Thus, the different cellular responses induced by androgens are caused by the activation of mAR or iAR. This evidence indicates that androgen elicits different signaling pathways activated by mAR or iAR in ER-positive breast cancer (Figure 1) and ER-negative breast cancer (Figure 2).

In breast cancer, EGF signaling regulates cancer progression through different signaling mechanisms such as cell motility and proliferation, regardless of the ER participation. Activation of the epidermal growth factor receptor (EGFR) induces the migration and proliferation of ER-negative breast cancer cells via Rac1/PI3K/Akt/p21-activated kinase 1 (PAK1) signaling and Src/MAPK/ERK/cyclin D1 signaling, respectively [48,49]. Additionally, in ER-positive MCF-7 cells, EGF-induced proliferation is mediated by PI3K/Akt and MEK/MAPK pathways, and this EGFR-driven proliferation is not influenced by ER [50]. In addition, protein kinase C (PKC)/MEK/ERK/early growth response protein 1 (EGR1) signaling is related to EGF-induced cell migration and invasion of MCF-7 cells [51]. Thus, the signaling pathway of EGF-induced cancer progression is parallel with the ER-driven cancer progression. In molecular apocrine tumors, in addition to the classical androgen-induced signaling transduction, an overexpression of HER2 associates with AR positivity that suggests a cross-talk between the AR and HER2 signaling pathways [52]. EGF could elicit the downstream molecules of the AR through transactivation. These two, in turn, act in synergy to stimulate cell proliferation [52,53,54]. Activation of EGFR triggers the formation of AR/Src complex and induces Src-dependent cell proliferation and migration [55]. Additionally, cross-talk between EGFR signals and the AR signal is also observed in ER-positive breast cancer [56].

The action of androgens can vary depending on the background of hormone receptors in different types of cancers. In ER-positive breast cancer MCF-7 cells, DHT induces cell growth via the ER and the ERK1/2-dependent signaling pathway [10]. Inhibition of the ER activity or downregulation of ERα reduces the DHT-induced proliferation [9]. The ER induces proliferation by recruiting other co-regulators, forming a transcriptional regulatory complex, or by modulating other transcription factors and developmental pathways [57]. The general ER signaling of cancer progression involves both genomic and non-genomic mechanisms [58]. Genomic signaling includes the activation of intracellular ERs upon ligand binding, causing their translocation to the nucleus. These nuclear ERs act as ligand-activated transcription factors, binding to the ER response elements on the target genes associated with various cellular processes [58]. Non-genomic estrogen effects engage the plasma membrane receptors like ER variants and the G protein-coupled estrogen receptor. Estrogen binding to these membrane receptors triggers non-genomic actions, characterized by intracellular second messengers, cAMP regulation, and protein kinase activation. These events lead to indirect alterations in gene expression through the binding of CREB to cAMP response elements on DNA [58].

The synthetic androgen R1881 rapidly stimulates the assembly of the AR/Src complex [59], leading to distinct cell progression in breast cancer. This complex promotes proliferation in ER-positive breast cancer cells through the PI3K/FAK/paxillin signaling pathway, while it enhances motility in ER-negative breast cancer cells through ERK1/2 signaling [60]. These findings highlight the role of the hormone receptor context in determining the cellular responses to androgens. Additionally, androgens exert diverse effects in different organs. For instance, they promote cell proliferation by upregulating fibroblast growth factor 8 (FGF8) and downregulating thrombospondin 1 (TSP1) expression in S115 mouse mammary tumor cells [61,62]. On the other hand, in bladder cancer cells, cell proliferation can be induced by DHT stimulation, even in the absence of AR expression, suggesting that the AR is not the only receptor responsible for this effect [63]. These observations suggest that the same effector, androgen, can induce different downstream effects in different organs.

Evidence revealed that the differential expression of the AR and ER plays a reverse role in cancer progression in ER-positive breast cancer [64]. The activation of the ER induced by estrogen in ER-positive/AR-negative cells or the AR induced by androgen in ER-negative/AR-positive cells promote tumor growth. In the absence of androgen, estrogen promotes the proliferation of ER-positive and AR-positive cells. However, when agonists activate the AR, it suppresses estrogen-driven cancer growth by altering the genomic distribution of the ER and essential coactivators such as p300 and steroid receptor coactivator 3 (SRC-3) [65]. This finding is controversial compared to other published studies, as it contradicts the notion that activation of the AR stimulates the growth of ER-positive and AR-positive breast cancer cells. This debatable issue may be attributed to the differences in the activation of mAR or iAR. Biased ligands such as DHT or testosterone can selectively activate distinct forms of the AR to modulate cell growth in ER-positive breast cancer. Therefore, conducting context-specific molecular studies that focus on biologically significant and clinically relevant pathways regulated by the AR can provide a better understanding of the contradictory effects of the AR in breast cancer.

## 3. Integrin-Related Signals in Breast Cancers

Integrins are heterodimeric structural proteins located in the plasma membrane. They facilitate cell–cell adhesion and mediate the interactions between the cells and ECM proteins upon ligand binding. Integrins are composed of α and β subunits, resulting in the formation of 24 different heterodimers. Integrins are a superfamily of cell adhesion receptors that bind to the ECM ligands on the cell surface and soluble ligands [66,67,68,69,70,71,72].

Integrins have been widely recognized and extensively studied for their role in mediating cancer cells’ proliferation, invasion, and metastasis, particularly in breast cancer. The survival rate for women diagnosed with metastatic breast cancer is generally lower than those with non-metastatic breast cancer. Numerous studies have provided evidence highlighting the significant involvement of integrins in breast cancer metastasis [73]. Outside-in signaling in integrin-mediated cell adhesion includes the activation of kinases in response to the adhesion process [74].

In ER-positive breast cancer cells, specific integrins, including integrin α1β1 and integrin α2β1, play a role in mediating cancer progression. In the early stage of breast cancer, integrin α1β1 is upregulated by ERβ to enhance cell adhesion for preventing cell migration [75]. On the other hand, in the late stage of breast cancer, the expression of integrin α2β1 is increased, along with the upregulation of mesenchymal markers such as vimentin, twist1, and N-cadherin. This is consistent with the progression of epithelial–mesenchymal transition (EMT) [76].

Compared to ER-positive cells, ER-negative cells exhibit the involvement of different integrin subtypes in cancer progression. Talin-1 (TLN1), a cytoplasmic adapter protein, mediates cell–cell adhesion by interacting with integrin β1 in breast cancer cells. It binds to and activates integrin β1, thereby regulating the dynamic formation of focal adhesions (FAs). This activation of integrin β1 promotes tumor metastasis through the involvement of the PI3K/AKT and FAK signaling pathways [77]. In addition, integrin α3β1 also regulates tumor metastasis through the PI3K/AKT/Brn-2 signaling pathway [78]. Breast cancer cells release many extracellular vesicles (EVs), facilitating intercellular communication in the tumor microenvironment and promoting metastasis. Galectin-3 assists in the export of integrin αvβ1 into these EVs. Integrin αvβ1 on the surface of EVs enables it to bind to fibronectin in the ECM, thereby supporting tumor metastasis [79]. Integrin β3 is essential for early and effective spontaneous breast cancer metastasis to the bone and soft tissue [80]. It promotes migration, expression of proteases, and trans-endothelial migration in vitro and enhances vascular dissemination in vivo. However, it is not essential for bone colonization in experimental metastasis assays. In addition, studies analyzing the differential gene expression in cohorts of breast cancer patients show a strong association between the high expression of integrin β3, early metastasis, and shorter disease-free survival in patients with ER-negative tumors [80]. In addition, the integrin αv is essential for efficient transforming growth factor-β (TGF-β)/Smad signaling and TGF-β-induced migration of breast cancer cells [81]. Additionally, knocking down integration with paclitaxel offers a more effective therapeutic option than combining cilengitide with paclitaxel [81]. These findings suggest that integrin αv could be a potential clinical therapy target for breast cancer treatment.

Integrin αvβ3, a member of Arg-Gly-Asp (RGD)-recognizing integrin, binds with ligands such as ECM proteins and growth factors [66,67,68,69,82]. Integrin αvβ3 expression is essential in developing bone metastasis in breast cancer [83]. Legumain (LGMN), an endo-lysosomal cysteine protease, has been found to exhibit a positive correlation with the metastatic progression of breast cancer. Motile breast cancer cells secrete legumain in its zymogen form. When the autocrine pro-legumain binds to the cell surface integrin αvβ3 through an RGD motif, it activates FAK-Src-ras homolog family member A (RhoA) signaling within cancer cells. This signaling pathway promotes cancer cell migration and invasion, and interestingly, this effect is independent of legumain’s protease activity. Angiogenesis involves recruiting new blood vessels and plays a crucial role in the metastatic pathway. It has been reported that the inhibition of integrin αvβ3 activity suppresses breast cancer cell angiogenesis [84,85]. In the presence of integrin αvβ3, neuropilin-1 minimally contributes to the vascular endothelial growth factor-induced angiogenic processes in vivo and in vitro [86].

X-ray diffraction has successfully demonstrated the crystal structure of the interaction between the RGD binding site of integrin αvβ3 and its ligand [87]. Integrin αvβ3 is expressed by high-growth endothelial cells, vascular smooth muscle cells, bone cells, and cancer cells [69]. Functions of this integrin in breast cancer cells include modulating cancer cell proliferation, tumor metastasis, and tumor-relevant angiogenesis [88]. Activation of integrin αvβ3 by DHT triggers phosphorylation of ERK1/2, which subsequently induces cell proliferation [43]. However, the effect of integrin αvβ3 on cell proliferation can be reversed by doxycycline treatment [89]. Extracellular connective tissue growth factor (CTGF) regulates cell proliferation and the migration of TNBC through the activation of integrin αvβ3. When CTGF binds to integrin αvβ3, it activates the FAK/Src/nuclear factor kappa-light-chain-enhancer of activated B cells (NF-κB) p65 signaling axis, which leads to the upregulation of glucose transporter 3 (Glut3) transcription [90]. Various integrin αvβ3 antagonists have been developed to inhibit cancer cell proliferation. The RGD motif is a potent target used in anti-tumor lipid microbubble (MB) therapy for breast cancer. Paclitaxel (PTX)@RGD-MBs, an MB with the outer lipid membrane containing an RGD motif and the inside space with PTX, has been designed to treat TNBC cells more effectively [91]. Ultrasonic targeted microbubble destruction (UTMD) is applied to improve MB penetration into the cell membrane, thereby increasing the concentration of PTX@RGD-MBs within the cells [91]. This PTX@RGD-MBs provides a better approach for diagnosing and treating TNBC cells. In another study by Ping Zhong et al., they used RGD-MBs to deliver a mertansine prodrug (cRGD-MMP) to suppress tumor growth. This study was conducted on nude mice with xenografts of MDA-MB-231 TNBC cells that overexpress integrin αvβ3 [92]. cRGD-MMP exhibited high potency against MDA-MB-231 cells in vitro, with a low half-maximal inhibitory concentration of 0.18 μM, which was 2.2-fold lower than that of the nontargeted MMP control. The tumor weights demonstrated that cRGD-MMP achieved an almost 2-fold higher tumor inhibition rate compared to the nontargeted MMP control. Consequently, cRGD-MMP displayed extended circulation time, enhanced tumor selectivity, and increased drug accumulation within the tumor.

## 4. Interaction between Androgen and Integrins

Hormones, including androgens, have demonstrated their ability to interact with integrins, thereby regulating various cellular processes (Figure 3). For example, androgen-stimulated AR signaling activates the AR/AR-associated protein 55 (ARA55)/FAK complex. This activation leads to the induction of collagen-integrin α2β1 gene expression, which is essential for promoting the migration of AR-mediated periosteum-derived progenitor cells (refers to integrin α2β1 in Table 1) [17]. Not only non-tumor cells but also tumor cells contribute to cell migration through the interaction of androgen and integrin. Migration is a fundamental process involved in cancer cell metastasis. Thus, androgen may interact with integrins to modulate cell invasion and metastasis.

Indeed, different subtypes of integrins have been implicated in regulating cell metastasis in various types of cancer (Figure 3A) (Table 1). In AR-negative prostate cancer cells, the integrin αvβ6 has been identified as a functionally active receptor for fibronectin and the latency-associated peptide of TGF-β1 [25]. Upon activation, this integrin selectively upregulates the expression of matrix metalloproteinase 2 (MMP2) and the activation of JNK1 in multiple prostate cancer cells in vitro, which plays a crucial role in promoting the osteolysis of bone metastasis (refers to integrin αvβ6 in Table 1) [25]. However, the expression of ARs in these AR-negative prostate cancer cells reduces the expression of integrin αvβ6, thereby leading to decreased cell interactions with the substrate and impaired migration ability (refers to integrin αvβ6 in Table 1) [29]. In contrast to the role of integrin αvβ6 in prostate cancer, integrin α6β1 is involved in the survival of castration-resistant prostate cancer (CRPC) cells in response to AR signaling [30]. Specifically, under both normoxic and hypoxic conditions, integrin α6β1 promotes AR-dependent cell survival by inducing the expression of Bnip3 [22]. Integrin α6β1 and Bnip3 have been identified to selectively promote the survival of CRPC cells on laminin through the induction of autophagy and mitophagy mechanisms (refers to integrin α6β1 in Table 1) [22]. After analyzing a prostate cancer tissue microarray patient cohort, CD151-associated integrin α3β1 and integrin α6β4 exhibited an inverse correlation with AR expression. This downregulation of the AR was associated with the reduced proliferation of cancer cells (refers to integrin α3β1 and integrin α6β4 in Table 1) [20]. SENL, a supercritical extract of neem leaves, has shown a similar anti-proliferative effect on prostate cancer. It exerts its suppressive effect on cell growth by inhibiting the expression of integrin β1 and the activation of AR and FAK, thereby impeding cancer cell proliferation [93]. This integrin β1 also mediates the AR-induced PI3K/AKT signaling pathway for suppressing the migration of hepatocellular carcinoma cells (refers to integrin β1 in Table 1) [14]. Studies have revealed that the AR can regulate the progression of prostate cancer cells by activating other receptors. One such receptor is the type 1 insulin-like growth factor receptor (IGF-IR) downstream of the AR signaling. IGF-IR can exert a pro-survival effect on prostate cancer by controlling the stability of the integrin α5β1 through a proteasomal pathway (refers to integrin α5β1 in Table 1) [13,21]. In LNCaP cells with a high expression of receptor activator of NF kappa-B ligand (RANKL), downregulation of the AP-4 transcription factor leads to the restoration of AR expression. In a 3D suspension culture model, both the AR and RANK regulate the expression of integrin α2 and enhance adhesion to collagen type 1 by activating the FAK and AKT signaling pathways (refers to integrin α2 in Table 1) [16]. Treatment of prostate cancer cells with DHT leads to the suppression of cell progression by altering the expression levels of key molecules involved in cell adhesion. Specifically, DHT treatment increases the expression of the anti-adhesion mucin 1 (MUC-1) while decreasing the expression of integrin α2β1 (refers to integrin α2β1 in Table 1) [18].

Androgen deprivation therapy (ADT) is a standard treatment for prostate cancer aimed at reducing the levels of androgen hormones, which are necessary for the growth of prostate cancer cells. By lowering the androgen levels, ADT inhibits the growth and spread of prostate cancer cells, thus helping to manage the disease. However, a highly aggressive subtype of prostate cancer known as neuroendocrine prostate cancer (NEPrCa) can emerge because of ADT. In NEPrCa cells, integrin αvβ3 and its effector, Nogo receptor NgR2, promote increased cell motility through the activation of RhoA (refers to integrin αvβ3 in Table 1) [27,28]. These findings may open the possibilities for developing new therapeutic strategies and identifying the risk factors in prostate cancer patients.

Integrin αvβ3 also regulates androgen-driven cell progression in breast cancer [9,10]. The interaction between integrin αvβ3 and DHT promotes cell proliferation induced by DHT in ER-negative breast cancer cells but not in ER-positive breast cancer cells (refers to integrin αvβ3 in Table 1) [9]. The effect of integrin αvβ3 on cell growth is mediated through the ERK1/2 signaling pathway [89]. In addition, DHT activates FAK leading to the reorganization of actin in breast cancer cells via the FAK, PI3K, and the Rac1 pathways [94].

FAK, a nonreceptor tyrosine kinase, generates signals upon activation to modulate important cell functions such as cell proliferation and migration. Activated FAK contributes to the activation of p66 Src homolog and collagen homolog (p66Shc), an isoform of Shc, promoting cell proliferation through rat sarcoma virus (Ras) and ERK1/2 [26,95,96]. The levels of p66Shc are higher in cancer cells than adjacent non-malignant cells in various cancer types, including breast, prostate, ovarian, thyroid, and colon carcinoma tissues [97]. Steroids-induced elevation of p66Shc and functional steroid receptors are required to proliferate prostate and ovarian cancer cells [98]. Integrin αvβ3 facilitates the recruitment and phosphorylation of β3-associated p66Shc, leading to the upregulation of vascular endothelial growth factor (VEGF) expression [26,98]. VEGF is an important protein involved in angiogenesis. Thus, integrin αvβ3 likely promotes tumor cell angiogenesis through the p66Shc/VEGF signaling pathway (refers to integrin αvβ3 in Table 1). Indeed, DHT binds to integrin αvβ3 and stimulates proliferation in ER-negative breast cancer cells, potentially through the direct or indirect activation of p66Shc phosphorylation via the VEGF signal pathway [26,96].

To our knowledge, no direct evidence shows how androgen interacts with integrin αvβ3. Here, based on our studies of DHT on breast cancer cells, we predict the relationship between integrin αvβ3 and DHT by the docking model. The docking conformation of DHT is shown in Figure 4. The DHT molecule employs its 17-OH to form a conventional hydrogen bond with the ASN215 residue of the β3 subunit (Figure 4C,D). The conformation of DHT places its steroid backbones underneath the cyclic RGD (cRGD)-binding site on the integrin (Figure 4B). A previous report indicated that the RGD-binding domain of integrin αvβ3 has three main binding pockets: a thyroid hormone pocket, a resveratrol pocket, and a steroid pocket [99,100]. This modeling of DHT is consistent with previous reports indicating that these steroid hormones bind to the lower region of the cRGD molecule. DHT exhibits metal interactions between the 17-OH and magnesium and forms hydrogen bonds with the amino acid residues of the β3 integrin subunit. These interactions are found in the aspartate (Asp) region of the cRGD peptide, which plays a crucial role in sex hormone binding. Thus, this predicted docking result could develop specific drugs for treating integrin αvβ3-dependent cancer progression.

## 5. Androgen, Integrin αvβ3, and PD-L1 Expression in Cancer Cells

PD-L1 is a transmembrane protein implicated in suppressing the adaptive immune system. It interacts with its receptor, PD-1, in activated T, B, and myeloid cells, to regulate their activation and inhibition. This PD-1/PD-L1 pathway has a subtle role in maintaining the peripheral T-lymphocyte tolerance and regulating inflammation [101]. Additionally, PD-L1 plays a significant role in various tumors by attenuating the host immune response to tumor cells. The PD-1/PD-L1 axis is responsible for cancer immune evasion and significantly impacts cancer therapy. An elevated level of PD-L1 is found in various cancers, including melanoma, lung, bladder, prostate, and breast [102,103]. The interaction of PD-L1 expressed on cancer cells and PD-1 expressed on T cells promotes cancer cell proliferation, EMT, invasion, and metastasis [102,104,105,106]. Silencing of PD-L1 not only reduces the proliferation and migration of cancer cells but also triggers apoptosis through intrinsic and extrinsic pathways [102,107]. PD-L1 expression in TNBC cells is higher than that in other subtypes of breast cancer cells [108,109]. In TNBC MDA-MB-231 cells, this increased level of PD-L1 is mediated by integrin αvβ3-induced BRAF/transforming growth factor-β-activated kinase 1 (TAK1)/ERK/E26 transformation-specific variant transcription factor (ETV4) signaling for inhibiting T cell function and evading T cell-mediated killing [110]. In addition, integrin αv expressed in non-small cell lung cancer (NSCLC) has a similar effect in preventing T cell-mediated killing [111]. This integrin activates autocrine TGF-β to regulate CD8 T cell immunity [111]. Patients with NSCLC receiving anti-PD-1 and anti-PD-L1 therapies have shown higher CD8+CD103+ tumor-infiltrating lymphocytes [111]. Thus, integrins may be biomarkers to predict the response to T cell-based cancer immunotherapies.

In various cancers, the expression of the AR is negatively correlated with the overall immune composition and is inversely associated with PD-L1 expression [112]. Furthermore, the AR is linked to a reduction in total macrophage infiltration [112]. However, this negative correlation between the AR and PD-L1 is not observed in aggressive or metastatic cancers [113,114,115]. Thus, these conflicting results might be due to the prior exposure of patients to anti-androgen hormone therapy or immunotherapy.

In TNBC MDA-MB-231 cells, the expression of the AR is not the highest among other breast cancer subtypes, but it still indirectly influences the face of PD-L1, which regulates cell progression. A positive correlation exists between the increased expression of hepatitis B X-interacting protein (HBXIP) and reduced macrophage infiltration with AR expression [112,116]. The transcription of PD-L1 is regulated by HBXIP-induced activation of E26 transformation-specific proto-oncogene 2 (ETS2) [116]. Additionally, HBXIP interacts with acetyltransferase p300 to induce acetylation of PD-L1 at the K270 site, thereby stabilizing the PD-L1 protein [116]. Depletion of HBXIP or aspirin treatment has been shown to attenuate PD-L1-induced tumor growth. These findings provide new insights into the mechanisms underlying the regulation of tumor PD-L1 and highlight the potential for targeting PD-L1 in breast cancer therapy [116]. Therefore, the expression of the AR is associated with specific immunological profiles in the breast cancer microenvironment at both the gene and protein expression levels.

## 6. Conclusions

Androgens induce closely related non-genomic and genomic signals in various cancers. Activation of the AR mediates cancer progression through the interaction of integrins. Indeed, the binding domain of the Asp residue of the cRGD ligand plays a pivotal role in the interactions between the hormones and the heterodimeric integrin αvβ3, as was predicted by molecular docking. In addition to integrin αvβ3, other subtypes of integrins also play essential roles in signal transduction, cell recruitment, and interactions with ECM proteins. Both in genomic and non-genomic actions, androgen-mediated integrins regulate nuclear transcription through different pathways, including classical nuclear receptor localization and changes in the phosphorylation of critical nuclear signaling molecules. PD-L1 expression affects cancer cell proliferation, interferes with chemotherapy, and negatively correlates with cancer progression. PD-L1-induced target genes are involved in cancer growth and metastasis. Integrins play a role in activating signal transduction pathways that can regulate the expression of PD-L1. Although the relationship between the expression of PD-L1 and the level of AR is controversial in cancers, PD-L1 probably controls androgen-mediated cancer behaviors through integrins. This regulation can occur through direct or indirect interactions with the AR. Understanding how androgens interact with integrins, such as integrin αvβ3, to induce PD-L1-mediated cellular processes in cancer can provide valuable insights for developing targeted therapeutic strategies.

## Figures and Tables

**Figure 1 cells-12-02126-f001:**
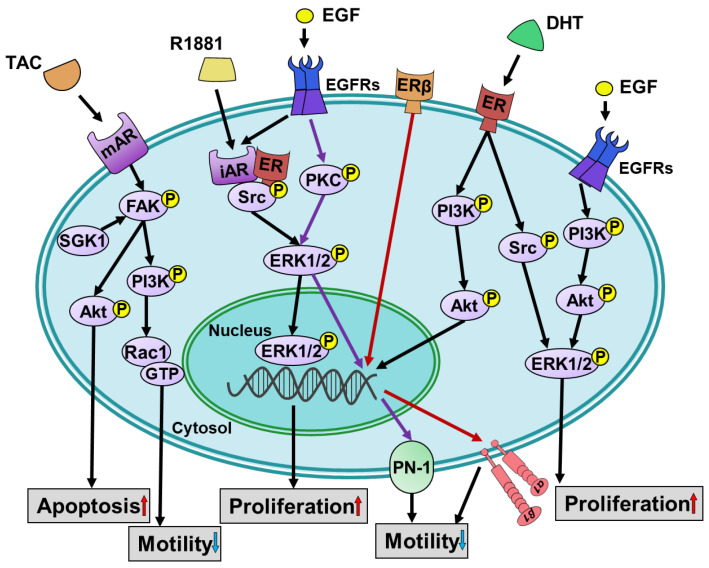
Schematically representing the signal transduction and biological activities induced by androgen in ER-positive breast cancer cells. Endogenous androgen, DHT, activates ER-mediated PI3K/Akt and ERK1/2 signaling pathways to regulate cell proliferation. Moreover, the process of ERK1/2-mediated cell proliferation is controlled by the activation of EGFRs stimulated by EGF, as well as the activation of iAR induced by the synthetic androgen, R1881. Upon EGFRs activation, it not only involves the formation of an association between iAR and Src complex but also triggers cell proliferation through PI3K/Akt/ERK1/2 signaling. Stimulation of mAR by TAC triggers the phosphorylation of FAK and Akt, resulting in enhanced cell apoptosis. In this mAR-mediated apoptosis, SGK1 plays a role in improving the apoptotic effect. Cell motility is downregulated upon mAR activation through the FAK/PI3K/Rac1 signaling pathway. In the absence of AR-driven signaling, EGFR induces PKC-mediated ERK1/2 activation and PN-1 expression to suppress cell motility. ERβ also inhibits cell motility through induction of integrin α1β1. The red arrows indicate Erβ/ integrin α1β1 pathway; The purple arrows indicate EGF/EGFR/PKC/ERK1/2/PN-1 pathway.

**Figure 2 cells-12-02126-f002:**
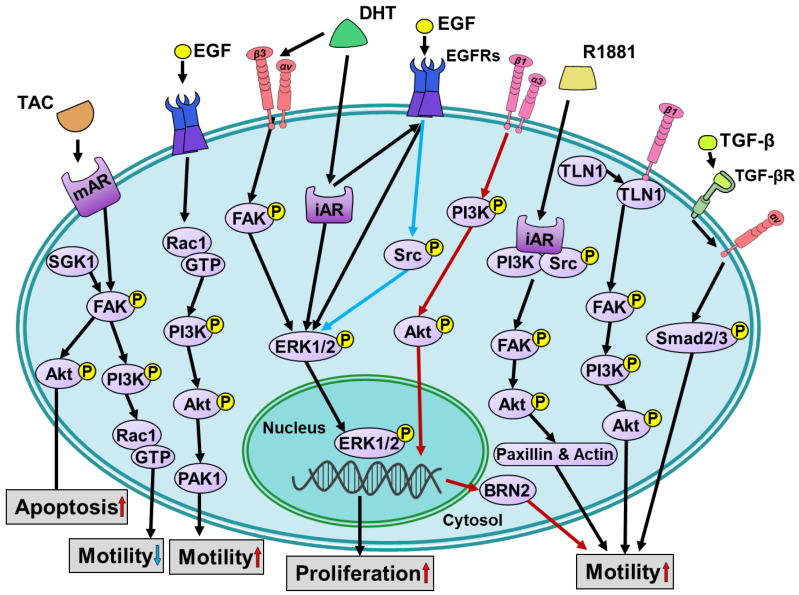
Schematically representing the signal transduction and biological activities induced by androgen in ER-negative breast cancer cells. Activation of iAR by DHT stimulates ERK1/2-mediated cell proliferation either directly or through cross-talk with EGFRs. Additionally, DHT binding to integrin αvβ3 activates the FAK/ERK1/2 signaling pathway, resulting in phosphorylated ERK1/2 translocation to the nucleus and subsequent cell proliferation. The association of iAR, PI3K, and Src induced by R1881 leads to cell motility through the FAK/Akt/paxillin signaling pathway. The function of mAR activated by TAC is similar to that in ER-positive breast cancer cells. Without AR-driven signaling, EGFR enhances cell motility and proliferation by triggered Rac1/PI3K/Akt/PAK1 signaling and Src/ERK1/2 signaling, respectively. Integrin α3β1 or β1 can direct activate PI3K/Akt signaling to regulate cell motility. After binding of TGF-βR with TGF-β, integrin αv phosphorylates Smad2/3 to increase cell motility. The red arrows indicate integrin α3β1/PI3K/Akt/BRN2 pathway; The blue arrows indicate EGF/EGFR/Src/ERK1/2 pathway.

**Figure 3 cells-12-02126-f003:**
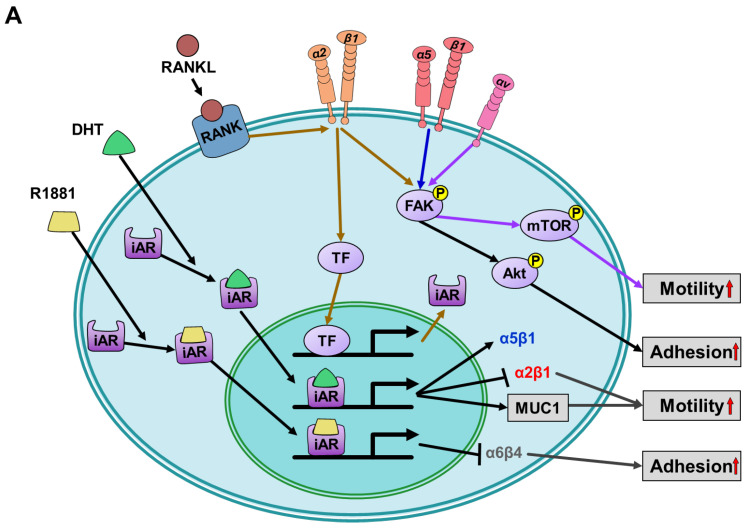
Diagram illustrating the androgen-mediated signal transduction via integrins and the resulting biological activities. (**A**) Integrin-induced cell motility and adhesion are mediated by AR. Activation of iAR by DHT or R1881 affects different subtypes of integrin expression to regulate cell adhesion and motility. RANK/RANKL promotes iAR expression through integrin α2β1 and induces integrin α2β1-involved cell adhesion via FAK/Akt signaling pathway. In AR-expressing cells, FAK signaling modulates integrin α5β1-induced cell adhesion and integrin αv-induced cell motility. (**B**) Integrin-induced cell growth and survival are mediated by AR. Activation of iAR by R1881 induces the expression of integrin α6β1, which in turn regulates cell survival through HIF-1α-promoted BNIP3 expression. Furthermore, iAR activity is up-regulated by integrin αvβ6, leading to increased survivin expression and enhanced cell growth. The activation of IGF-IR suppresses the degradation of integrin α5β1 through endocytosis, allowing this remaining integrin α5β1 to induce cell growth. The brown arrows indicate RANK/integrin α2β1/ transcription factor (TF)/iAR and RANK/integrin α2β1/FAK/Akt pathways; The dark blue arrows indicate integrin α5β1/FAK/Akt pathway; The purple arrows indicate integrin αv/FAK/mTOR pathway; The light blue arrows indicate IGF-1R/ integrin α5β1 pathway; The red arrows indicate R1881/iAR/integrin α6β1 pathway; The green arrows indicate integrin αvβ6/JNK/iAR (activity)/survivin pathway; The orange arrows indicate integrin α6β1/HIF-1α/BNIP3 pathway.

**Figure 4 cells-12-02126-f004:**
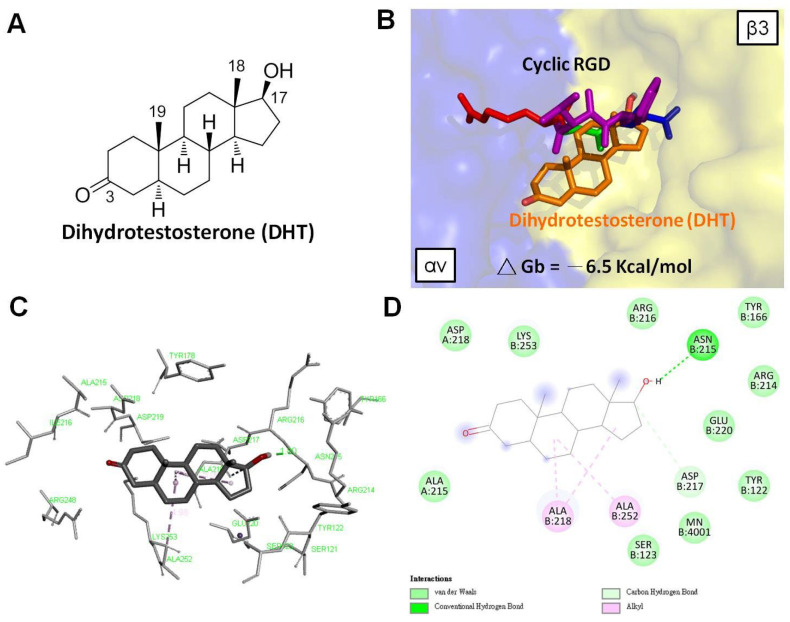
Predicted docking poses of DHT bound at the cRGD−binding site of integrin αvβ3. The generation of docking poses and further scoring were done by AutoDock 4 and PyMOL, as previously described [99]. (**A**) The two−dimensional structure of DHT. (**B**) Close−up of DHT binding mode, superimposed with cRGD peptide within the integrin αvβ3. (**C**,**D**) The amino acid residues of integrin αvβ3 interact with the DHT molecule.

**Table 1 cells-12-02126-t001:** Androgen receptors and their functions in non-cancer/cancer cells.

Receptor	Cancer Type	Functions	Reference
AR	Prostate cancer	1. To form a ligand–AR complex and to control gene expression 2. To stimulate the proliferation of prostate cancer cells	[12]
ERα	ER-positive breast cancer	To stimulate the proliferation of ER-positive breast cancer cells	[9]
Integrin αv	Prostate cancer	To regulate tumor cell migration and growth	[13]
Integrin β1	Hepatocellular carcinoma	To induce cell adhesion through PI3K/AKT signaling pathway	[14]
Integrin β1C	Prostate cancer	To be correlated with prostate cancer progression	[15]
Integrin α2	Prostate cancer	To regulate metastasis mediated by adhesion to ColI through RANKL/RANK signaling	[16]
Integrin α2β1	Periosteum-derived progenitor cells	To involve cancer cell migration	[17]
	Prostate cancer	To regulate cancer progression	[18]
		To be controlled its expression by AR	[19]
Integrin α3	Prostate cancer	To regulate tumor cell growth	[13]
Integrin α3β1	Prostate cancer	To repress cell proliferation and EMT in prostate cancer by CD151	[20]
Integrin α5β1	Prostate cancer	To promote prostate cancer growth	[21]
Integrin α6β1	Prostate cancer	To promote the survival of CRPC cells selectively on laminin through the induction of autophagy and mitophagy	[22]
Integrin α6β4	Prostate cancer	To be involved in invasion	[23]
		To promote the survival of cancer cells	[24]
		To repress cell proliferation and EMT by CD151	[20]
		To promote an osteolytic program in cancer cells by upregulating MMP2	[25]
Integrin αvβ3	ER-negative breast cancer	To stimulate proliferation of ER-negative breast cancer cells	[10]
	Breast cancer	To regulate cell proliferation	[9]
	Prostate cancer	To regulate cell proliferation through the p66Shc/VEGF pathway	[26]
		To induce neuroendocrine differentiation through NOGO receptor NgR2	[27]
	Neuroendocrine prostate cancer	To promote cancer metastasis	[28]
Integrin αvβ6	Prostate cancer	To promote an osteolytic program in cancer cells by upregulating MMP2	[25]
		To induce cell adhesion and migration	[29]
	Castration-resistant prostate cancer	To promote cancer cell survival	[24]
		To promote survival and resistance to PI3K inhibition	[30]

## Data Availability

Not applicable.

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
