# Peer review of "Integrins and Actions of Androgen in Breast Cancer"

_cells, 2023, doi:10.3390/cells12172126_

Round 1
Reviewer 1 Report
Integrins and Actions of Androgen in Breast Cancer
This review focuses on the role of integrins and androgen signaling in breast cancer. As the topic is not extensively studied and reviewed, this review will help readers to have a general understanding of the related field. Overall, the review is well written.
I have following concerns that need to be addressed before the acceptance of the review for publication.
1. The authors should provide a general picture regarding ER signaling as it is a central topic of the review. For example, intracellular ER is the mostly important ER in terms of the biological function and the most established function of ER is the regulation of gene expression by migrating to nucleus and binding to DNA. This is not discussed in the review.
2. Similarly, EGF-EGFR signaling to Erk, PI3K and other signaling molecules are mostly independent of Androgen and integrin (they surely cross talk as discussed in the review), also independently of ER. This aspect is also lacking in the review.
3. Especially, Fig. 1 and 2 missed most important signaling pathways of ER, EGFR, and integrin, thus need to be reworked.
4. As shown in Fig. 1&2, TCA induced mAR signaling pathway has no interaction with ER, why it is different in ER-positive and negative breast cancer. Why integrin only interacts with this signaling pathway (phosphorylated FAK) in ER negative breast cancer.
5. Diagrams similar to Fig. 1&2 also should provided regarding Integrin signaling, and the interaction between AR signaling and various integrins. The current depiction in Fig.2 regarding this is too simplified.
English is fine in general, but some moderate editing is necessary.
Reviewer 2 Report
No major issues found.
Minor issues
· Line 47: I would provide a sentence here elaborating on why this is true that there is a higher incidence of recurrence and metastasis in TNBC.
· Line 189: Include citations of the studies providing evidence of integrin involvement in breast cancer metastasis.
· Line 257: For reference 65, provide results from the study conducted on the mice with TNBC cells overexpressing integrin αvβ3
Grammar
· Line 34: change “in actions by” to “in action with”
· Line 54: change “Comparison of” to “Compared to”
· Line 55: change “cells, indicating the different levels” to “cells. This indicates that different levels”
· Line 65: change “to play a vital role in the pathogenetic role” to “to play a vital pathogenetic role”
· Line 68: change “Besides, how PD-L1 is involved in androgen..” to “Additionally, the involvement of PD-L1 in androgen…”
· Line 118: change “These evidence indicate” to “This evidence indicates”
· Line 282: change “integrin α6β1 did involve in” to “integrin α6β1 is involved in”
· Line 289: change “exhibited an inversely correlated with” to “exhibited an inverse correlation with”
· Line 329: change “Shc, and promotes” to “Shc, promoting”
Only minor edits required.
Round 2
Reviewer 1 Report
The authors have addressed my concerns and the revised MS is acceptable for publication.
English is fine in general and only minor issues.